# Control of Dual-Output DC/DC Converters Using Duty Cycle and Frequency

**Yoshinori Matsushita [1,2,*], Toshihiko Noguchi [1], Kazuki Shimizu [3], Noritaka Taguchi [2] and Makoto Ishii [2]**

[1]   Graduate School of Science and Technology, Shizuoka University, Hamamatsu 432-8561, Japan; noguchi.toshihiko@shizuoka.ac.jp

[2]   Yazaki Research and Technology Center, YAZAKI Corporation, Susono, Shizuoka 410-1194, Japan; noritaka.taguchi@jp.yazaki.com (N.T.); makoto.ishii@jp.yazaki.com (M.I.)

[3]   Graduate School of Integrated Science and Technology, Shizuoka University, Hamamatsu 432-8561, Japan; shimizu.kazuki.16@shizuoka.ac.jp

*   Correspondence: yoshinori.matsushita@jp.yazaki.com

**Abstract:** As part of the integration process of the auxiliary power systems of electric vehicles, plug-in hybrid vehicles and fuel cell vehicles, this study proposes a method to control two different voltage types using two control factors of the rectangular alternating waveforms contained in DC/DC converters, namely the duty cycle and frequency. A prototype circuit consisting of an H-bridge inverter, a transformer, two series resonant filters and two diode bridge circuits was constructed. The H-bridge inverter was connected to the primary side of the transformer and the diode bridge rectifier circuit was connected to the secondary side in parallel. Series resonant filters were inserted between one of the diode bridge circuits and the transformer. Thereafter, the proposed control method was applied to the transformer voltage of the prototype circuit. Although the circuit operation became complex owing to the circulating current flowing between the ground (GND) of the two output circuits, it exhibited ideal static and dynamic characteristics, thereby confirming the possibility of controlling two voltages with the duty cycle and frequency control factors. The results of the efficiency evaluation and loss analysis demonstrated a minimum efficiency of 68.3% and a maximum efficiency of 88.9%. As the output power of the circuit containing the resonant filters increased, the current peak value increased and the circuit became less efficient.

**Keywords:** dual output; DC/DC converter; pulse width and frequency control; auxiliary power source; duty cycle; frequency

## 1. Introduction

### 1.1. Research Motivations

Considering that the transportation field is estimated to be responsible for 26% of all $CO_2$ emissions globally [1], the reduction of $CO_2$ emissions from vehicles is an effective measure to curb global warming. In an attempt to reduce $CO_2$ emissions, the vehicle market is switching from gasoline cars to electric vehicles (EVs), plug-in hybrid vehicles (PHVs) and fuel cell vehicles (FCVs), which are equipped with high-voltage sources that enable motor driving. The market for such vehicles is expected to grow continually in the coming years [2]. The auxiliary power sources of EVs, PHVs and FCVs use isolated DC/DC converters with input that provides high-voltage power for driving [3,4]. With the electrification of conventional functions (drive by wire) and the sophistication of internal environments, the number and types of auxiliary loads are increasing continually. That is, the power

capacity required for the auxiliary power supply is increasing and, depending on the use conditions of the car, the power consumption of the auxiliary load may have a significant impact on the total vehicle power consumption [5,6]. One factor that increases the power consumption is the additional conduction losses caused by the large current that flows when a heavy load is driven with a 12 V power supply. An option for reducing these conduction losses is to set the power supply voltage of the auxiliary system to 48 V and to reduce the current value [7,8]. However, owing to the costs, part supplies and maintenance service involved, it is difficult to make all loads for a 12 V power supply compatible with a 48 V power supply and, as a result, the demand for loads requiring a 12 V power supply remains. For this reason, the auxiliary power supply requires a dual system that supports both 48 V and 12 V.

The configuration known as the 48-V mild hybrid, standardized as LV148 in Europe, supplies a dual system of 48 V and 12 V by generating a direct current of 48 V in the alternator of engine-powered vehicles and producing a 12 V power supply with a non-isolated bidirectional DC/DC converter of 48 V and 12 V [9]. However, the fact that two power conversions are required to generate 12 V makes this system inefficient. Moreover, the system contains two sets of power supply equipment, making it expensive and bulky.

*1.2. Literature Review*

To overcome these issues, numerous studies have been conducted on single-input, dual-output DC/DC converters that input high voltages for driving and output 48 V and 12 V, as well as bidirectional multiport converters that make use of a power supply from renewable energy sources installed in cars and V2X [10–16]. On the other hand, studies have been carried out on mastering GaN devices since GaN devices provide a high switching speed that enable operations at higher frequencies than those of conventional Si and SiC devices, which makes power converters more compact and efficient [17–21].

The studies of the auxiliary power sources of cars have exhibited functions such as bidirectional power transmission and four or more inputs/outputs. However, as a result of the multiple functions, the number of switching devices sending control signals has increased, thereby increasing the total wiring length of the control signals. As control signals are vulnerable to noise, they are incompatible with the high frequencies resulting from the use of GaN devices or increases in the power capacity of auxiliary equipment. Furthermore, when the total wiring length of the control signal increases, the layout becomes more complex to prevent induction noise from the power section. However, few studies have focused on the issue.

*1.3. Contribution*

To attenuate this problem, the authors have proposed a method for controlling single-input, dual-output DC/DC converters as integrated auxiliary power sources for EVs, PHVs and FCVs, with fewer control signals and a shorter wiring length [16]. The method has novel characteristics switching devices on input port are able to control two output voltages, which is not shared by References [10–15]. However, Reference [16] only had demonstrated the efficacy of the method with simulation. In this paper, therefore, the validity of the method is verified using an actual circuit with 1 kW output.

## 2. Overview of Proposed Control Method and Main Circuit Configuration

*2.1. Overview of Proposed Control Method and Main Circuit Configuration*

Figure 1 presents the concept of the proposed control method. The two output voltages $V_{out1}$ and $V_{out2}$ are controlled by two control factors, namely the duty cycle and frequency (period) with rectangular alternating voltage. This control method requires the main circuit to contain waveforms with a duty cycle and frequency, as well as elements with output voltages that vary according to the changes in the duty cycle and frequency. Therefore, in this study, the isolated DC/DC converter

depicted in Figure 2 was adopted as a simple main circuit to validate the proposed control method. The isolation approach used consists of a transformer, the primary side of which is an input capacitor ($C_{in}$) and an H-bridge inverter that is formed by the switching devices S1, S2, S3 and S4. The gate signals G1, G2, G3 and G4 are input to each switch, respectively. G1 to G4 generate a rectangular alternating voltage with the duty cycle and frequency control factors and this voltage is applied to the primary side of the transformer. On the secondary side of the transformer, a rectifier circuit of output voltage $V_{out1}$ consisting of diodes D11, D12, D13, D14, $L_{out1}$ and $C_{out1}$ and, similarly, a rectifier circuit of output voltage $V_{out2}$ consisting of D21, D22, D23, D24, $L_{out2}$ and $C_{out2}$ are connected in parallel. A series of resonant filters consisting of $L_{srA}$, $C_{srA}$, $L_{srB}$ and $C_{srB}$ are inserted between the latter diode bridge circuit and the transformer. This series resonant filter changes the impedance to an arbitrary value by operating the frequency and thereby controls the output voltage. To maintain the symmetry of the operation, the parameters of the two series resonant filters must have the same values. Impedance changes caused by frequency affect the impedances of both the series resonant filter and the subsequent smoothing filter. In this case, if $L_{sr1} = L_{sr2} = L_{sr}$ and $C_{sr1} = C_{sr2} = C_{sr}$, the impedance of the series resonant filter ($Z_{sr}$) to the frequency of the transformer voltage ($f_{tx}$) and the impedance following the smoothing filter of the $V_{out2}$ side ($Z_{sm2}$) can respectively be expressed as

$$Z_{sr}(f_{tx}) = j\left(\omega L_{sr} - \frac{1}{\omega C_{sr}}\right), \tag{1}$$

$$Z_{sm2}(f_{tx}) = \frac{R_{out2}}{1 + (2\omega)^2 R_{out2}{}^2 C_{out2}{}^2} + j\left\{(2\omega)L_{out2} - \frac{(2\omega)R_{out2}{}^2 C_{out2}}{1 + (2\omega)^2 R_{out2}{}^2 C_{out2}{}^2}\right\}, \tag{2}$$

where $\omega = 2\pi f_{tx}$ and $R_{out2}$ is the load resistance value of the $V_{out2}$-side circuit. Considering that the frequency of the waveforms doubles after the diode bridge full-wave rectification, the frequency in $Z_{sm2}$ is doubled. With these values, $Z_{out2}$, which is the impedance of the $V_{out2}$-side circuit for $f_{tx}$, can be approximated as

$$\left|Z_{out2}(f_{tx})\right| = 2|Z_{sr}| + |Z_{sm2}|. \tag{3}$$

If the condition is satisfied whereby $|Z_{out2}(f_{tx})|$ decreases or increases monotonically within the operating frequency range, $f_{tx}$ and $|Z_{out2}|$ exhibit a one-to-one relationship and $V_{out2}$ can be controlled by $f_{tx}$. Moreover, as the side circuit $V_{out1}$ is a typical diode bridge rectifier circuit, $V_{out1}$ can be changed by the duty cycle of the secondary side voltage of the transformer ($D_{tx}$). Therefore, $V_{out1}$ is mainly controllable by $D_{tx}$ and $V_{out2}$ is mainly controllable by $f_{tx}$ but changes in $D_{tx}$ affect $V_{out1}$ as well as $V_{out2}$, whereas changes in $f_{tx}$ affect $V_{out2}$ as well as $V_{out1}$. Thus, it is not possible to control $V_{out1}$ and $V_{out2}$ independently using $D_{tx}$ and $f_{tx}$, respectively. However, by adding a gap in the response speed of both output voltages, it is possible to prevent interference in the control by $D_{tx}$ and $f_{tx}$ and to adjust both output voltages to the target values.

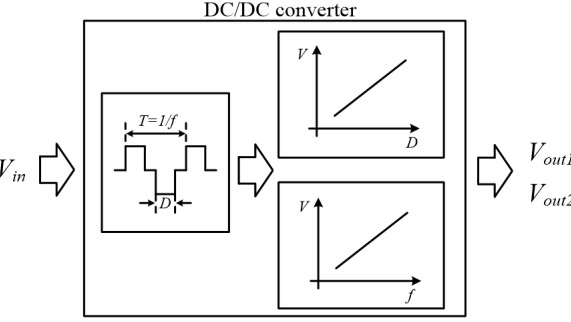

**Figure 1.** Concept of dual-output control.

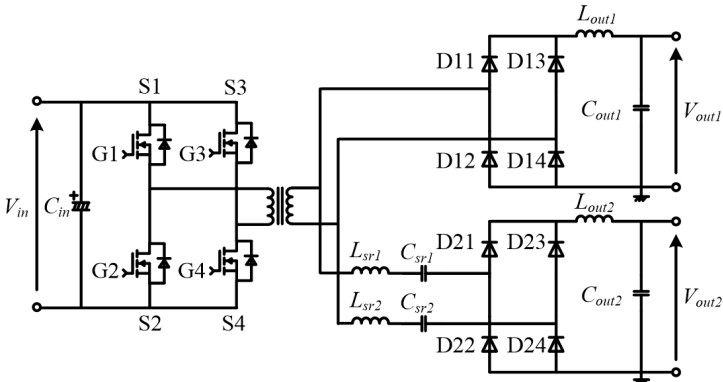

**Figure 2.** Circuit configuration for applying proposed control method.

Although the proposed control method requires the incorporation of a circuit such as a series resonant filter to link the frequency and control target, it is possible to control two output voltages using only the switching devices necessary to generate an AC rectangular waveform voltage (that is, without adding new switching devices for dual-output control). This reduces the layout restrictions to prevent inductive noise from the large current part of the main circuit, resulting in greater flexibility in the circuit design. 

*2.2. Operating Principles of Proposed Control Method*

Figure 3 presents a block diagram of the proposed control method. The calculation is conducted by a Field Programmable Gate Array (FPGA) and the signals input into the FPGA are the feedback signals $V_{fb1}$ and $V_{fb2}$, which are obtained by dividing by $V_{out1}$ and $V_{out2}$, respectively. $V_{fb1}$ and $V_{fb2}$ are input into the FPGA via the AD converter and the differences from the respective target values $V_{out1}^*$ and $V_{out2}^*$ are input into the PI calculation part (PI$_1$ and PI$_2$). The output signal from PI$_1$ contains the information of the phase shift amount $\delta$ ($\alpha_\delta$) required by the pulse width control to define $D_{tx}$, whereas the output signal from PI$_2$ contains the information necessary to define $f_{tx}$ ($\alpha_T$). These signals are input into the pulse width and frequency control (PWFC) section and are converted into the switching signals G1' to G4', which contain the information of $D_{tx}$ and $f_{tx}$ needed for the transformer voltage to control $V_{out1}$ and $V_{out2}$. Thereafter, a signal with a certain dead time provided to G1' to G4' is output from the FPGA and it is subsequently output to the gates of S1 to S4 via the isolated gate driver (G1 to G4).

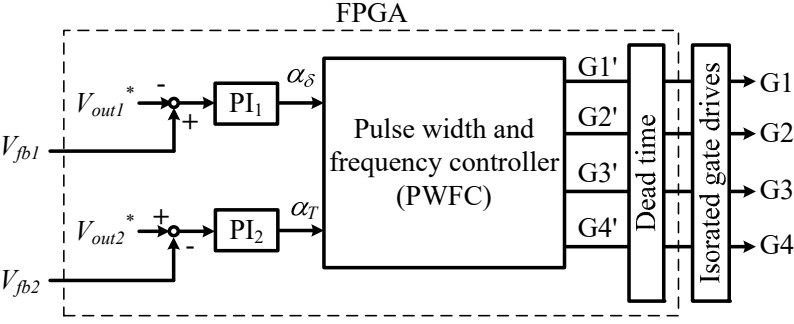

**Figure 3.** Control block diagram of proposed circuit.

Figure [4] shows the PWFC principle of switching the signal generation. The output signals from the PWFC part are G1′, with a variable frequency and a constant duty cycle of 0.5; G3′, which is phase-shifted from G1′ by $\delta$; and G2′ and G4′, which are inverted versions of G1′ and G3′, respectively. A sawtooth wave with an upper limit of the crest value set by $\alpha_T$ is generated by a counter inside the PWFC to generate these signals. This sawtooth wave counter decreases from the upper limit defined by $\alpha_T$ and when the counter reaches zero it is set to the upper limit again. As the decrease ratio of the counter corresponds to the clock period ($T_{CLK}$) of the FPGA and remains constant, the period of the sawtooth wave $T_{saw}$ (frequency $f_{saw}$) is defined by $\alpha_T$. This is the period $T_{tx}$ (frequency $f_{tx}$) that is necessary for the control. Furthermore, $f_{tx}$ is the switching frequency of G1′ to G4′ and is expressed as

$$f_{tx} = \frac{1}{\alpha_T T_{CLK}}, \tag{4}$$

in which the possible range for $\alpha_T$ is

$$\frac{1}{f_{MAX} T_{CLK}} \leq \alpha_T \leq \frac{1}{f_{min} T_{CLK}}. \tag{5}$$

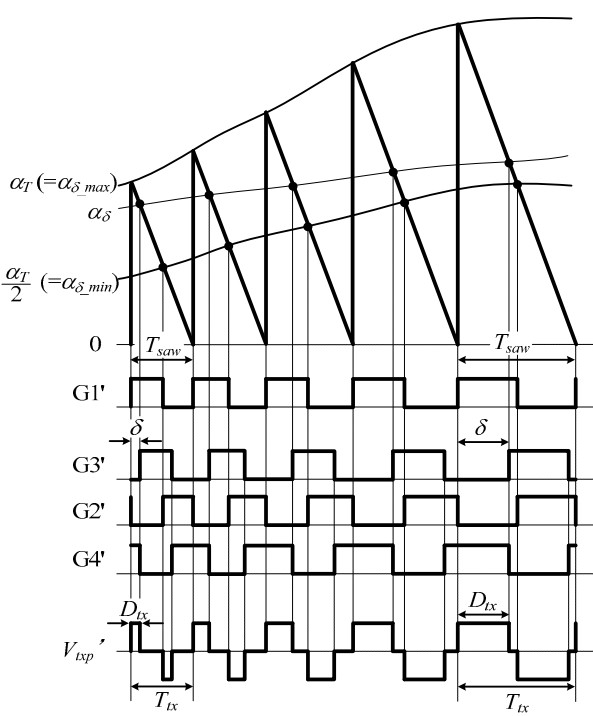

**Figure 4.** Principle of Field Programmable Gate Array PWFC function.

In the above, $f_{MAX}$ and $f_{min}$ are the maximum and minimum values of the switching frequency, respectively. By turning a signal on when the counter reaches the upper limit ($\alpha_T$) and off when the counter reaches half of the upper limit ($\alpha_{T/2}$), it is possible to generate rectangular waveforms with a frequency corresponding to $\alpha_T$ and a duty cycle of 0.5. This signal is designated as the G1′ signal.

While G1′ is generated, G3′ is generated by a comparison of $\alpha_\delta$ and the sawtooth wave counter. By turning G3′ on when the sawtooth waves become smaller than $\alpha_\delta$ and turning it off when the same amount of time since G1′ was turned on has elapsed, G3′ becomes a signal that is phase-shifted from G1′ by $\delta$. Because the range of $\delta$ where the control holds is between 0° and 180°, the possible range of $\alpha_\delta$ is

$$\frac{\alpha_T}{2} \leq \alpha_\delta \leq \alpha_T. \tag{6}$$

The switching signals G1′ to G4′ are obtained by expressing the inverted signals of G1′ and G3′ as G2′ and G4′, respectively. As illustrated in Figure 4, $V_{txp}$,′ which is on the primary side voltage of the transformer generated by G1′ to G4′, $D_{tx}$ and $T_{tx}$ (=1/$f_{tx}$) change according to the input values of $\alpha_T$ and $\alpha_\delta$. However, although $f_{tx}$ is determined uniquely by $\alpha_T$, $D_{tx}$ is not determined uniquely by $\alpha_\delta$, because the sawtooth shape determined by $\alpha_T$ is calculated by comparison with the counter.

## 3. Verification of Proposed Method Using Actual Circuit

### 3.1. Circuit Specifications

The specifications, circuit diagram and outer appearance of the actual circuit created to verify the proposed control method are presented in Table 1 and Figure 5, respectively. The output power of the $V_{out1}$ and $V_{out2}$ sides are expressed as $P_{out1}$ and $P_{out2}$, respectively. The maximum value of the total output power, $P_{out1} + P_{out2}$, was set to 1 kW and the maximum value of $P_{out2}$ was set to 500 W. The input voltage $V_{in}$ was set to 300 V, whereas the target values of the output voltages $V_{out1}$ and $V_{out2}$ were set to 48 V and 12 V, respectively. The minimum and maximum switching frequencies, $f_{min}$ and $f_{MAX}$, were set to 50 kHz and 100 kHz, respectively and the FPGA used was a XC7K70T-1FBG484C with a clock frequency ($f_{CLK}$) of 200 MHz ($T_{CLK}$ = 5 ns). When substituting the values of $f_{min}$, $f_{MAX}$ and $T_{CLK}$ into Equation (5), $\alpha_T$ can take values between 2000 and 4000. As $\alpha_T$ is an integer, $f_{tx}$ and $|Z_{out2}|$ could take 2001 possible values under the verification control conditions. Proportional gains and time constants of PI$_1$ and PI$_2$ were $K_1$ = 2.5, $K_2$ = 0.5, $\tau_1$ =3 μs, $\tau_2$ = 8 μs, respectively.

**Table 1.** Specifications of experimental circuit.

| Parameter | Value |
|---|---|
| $P_{out1} + P_{out2}$ (MAX) | 1 kW |
| $P_{out2}$ (MAX) | 500 W |
| $V_{in}$ | 300 V |
| $V_{out1}$ | 48 V |
| $V_{out2}$ | 12 V |
| $f_{swmin}$ | 50 kHz |
| $f_{swMAX}$ | 100 kHz |
| FPGA | XC7K70T-1FBG484C |
| $f_{CLK}$ ($T_{CLK}$) | 200 MHz (5 ns) |
| $K_1$ | 2.5 |
| $K_2$ | 0.5 |
| $\tau_1$ | 3 μs |
| $\tau_2$ | 8 μs |
| $C_{in}$ | 330 μF |
| S1, S2, S3, S4 | SCT3030AL (ROHM) |
| D11, D12, D13, D14, D21, D22, D23, D24 | FFSH4065A (ON Semiconductor) |
| Transformer turn ratio | N1:N2 = 20:5 |
| $L_{sr}$ | 4.5 μH |
| $C_{sr}$ | 560 nF |
| $L_{out1}$ | 7.3 μH |
| $L_{out2}$ | 7.0 μH |
| $C_{out1}$ | 44 μF |
| $C_{out2}$ | 188 μF |
| $f_0$ | 100.3 kHz |

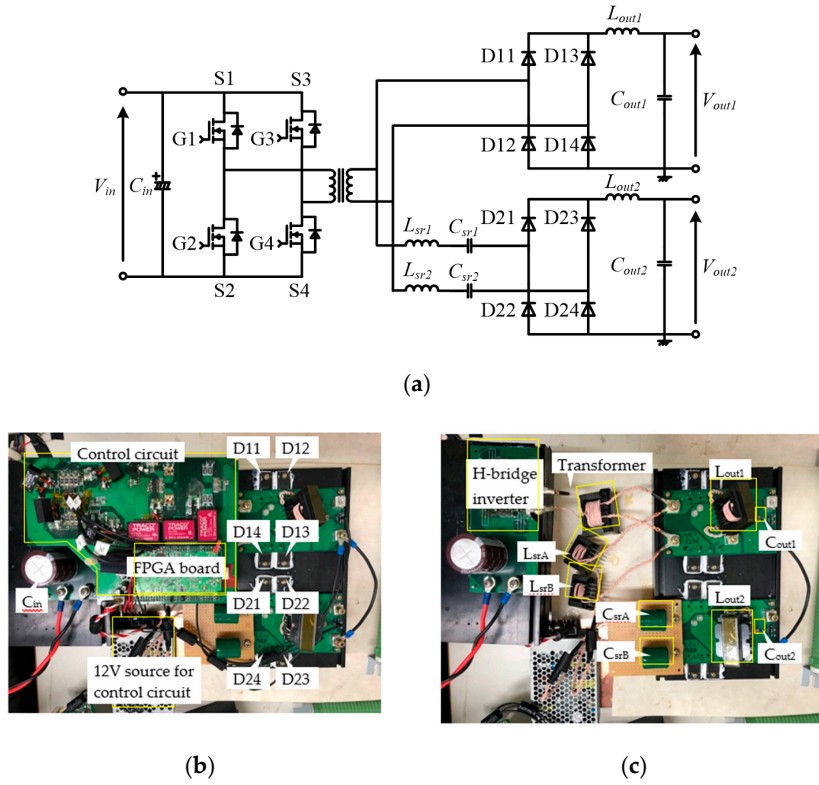

**Figure 5.** Experimental circuit diagram and setup: (**a**) Main circuit diagram, (**b**) Main circuit with control circuit; (**c**) Main circuit without control circuit.

The input capacitor $C_{in}$ was an aluminum electrolytic capacitor of 330 µF, the switching devices of the primary side H-bridge circuit (S1 to S4) were SiC MOSFET (SCT3030AL, ROHM) and the eight diodes of the secondary side diode bridge were SiC Schottky barrier diodes (FFSH4065A, ON Semiconductor). Because this circuit was designed to verify the proposed control method, the breakdown voltage and current capacity of these devices were substantially larger than necessary. The turn ratio of the transformer was 20:5. The parameters of the elements of the series resonant filter, $L_{sr}$ and $C_{sr}$, were 4.5 µH and 560 nF, respectively. The output smoothing inductors were $L_{out1}$ = 7.3 µH and $L_{out2}$ = 7.0 µH, the output capacitor Cout1 was 88 µF with four ceramic capacitors of 22 µF in parallel and $C_{out2}$ was 188 µF with four ceramic capacitors of 47 µF in parallel. The values of $C_{in}$, $C_{out1}$ and $C_{out2}$ are nominal values and the values of $L_{sr}$, $L_{out1}$ and $L_{out2}$ are calculated by using the following equation:

$$ L = \frac{\frac{di}{dt}}{V}. \tag{7} $$

In the above, *di/dt* is the rate of current change and *V* is the voltage across the inductor. Both *di/dt* and *V* are obtained by the experimental measurement. The data range both parameters are constant is used for the calculation.

The resonant frequency of the series resonant filter, $f_0$, could be calculated as 100.3 kHz according to the following equation:

$$ f_0 = \frac{1}{2\pi \sqrt{L_{sr}C_{sr}}} . \tag{8} $$

Each value of Table 1 was substituted into Equations (1) and (2) to calculate $Z_{sr}$ and $Z_{sm}$, which were then substituted into Equation (3). The result is presented in Figure 6. In this case, it was assumed that Rout2 → ∞. As indicated in Figure 6, the resonant frequency of the $V_{out2}$-side circuit was approximately 100 kHz. It can be observed that this value was the resonant frequency of the resonant filter and it was not affected by the output smoothing LC filter. This is because the resonant frequency of the

output smoothing LC filter ($L_{out2}$, $C_{out2}$) was approximately 4.4 kHz, which was far from 100.3 kHz. Moreover, the $Q$ value of the output smoothing LC filter was less than 10% of the $Q$ value of the series resonant filter (assuming that both filters had the same line resistance). As the resonant frequency required for control was approximately 100 kHz, in the operating frequency range between 50 kHz and 100 kHz, the impedance of the $V_{out2}$-side circuit decreased monotonically as the frequency of the transformer voltage ($f_{tx}$) increased, which enabled $V_{out2}$ to be controlled by $f_{tx}$, as mentioned previously. The reason for the monotonic decrease instead of an increase is that the volume of $L_{out1}$ and $L_{out2}$ could be reduced with higher frequency when a large current passed ($Z_{out2}$ was small).

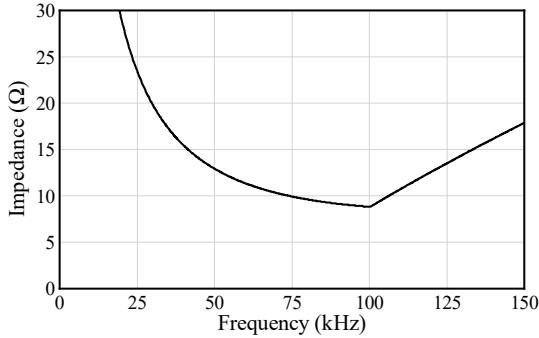

**Figure 6.** $Z_{out2}$–f characteristics of $V_{out2}$-side circuit.

As described above, the change rate of impedance by the frequency change and resonant frequency can be calculated using Equation (3) and Figure 6. The resonant frequency is the upper or lower limit of the operating frequency. Furthermore, the number of possible frequency values is defined by Equation (5). If the operating frequency is too high, the core losses and switching losses increase, whereas if it is too low, the volume of the magnetic core increases. Moreover, if the change rate of the impedance relative to the frequency change is excessively high and an FPGA without a high clock frequency is used, the impedance values that it can take are not continuous and the control resolution decreases. However, if the change rate of the impedance relative to the frequency change is excessively small, the control resolution increases but if the operating frequency range is not expanded, the power range that can be controlled by the $V_{out2}$-side circuit becomes narrower. Therefore, for the proposed control method to work as intended, it is necessary to select the most appropriate resonant frequency, operating frequency range, frequency–impedance characteristics and FPGA.

*3.2. Static Characteristics*

Figure 7 presents the output voltages $V_{out1}$ and $V_{out2}$ and output currents $I_{out1}$ and $I_{out2}$, with a total output of 1,037 W ($P_{out1}$ = 461 W and $P_{out2}$ = 576 W), from power-up until reaching a steady state. The measurement devices used in the actual verification are listed in Table 2.

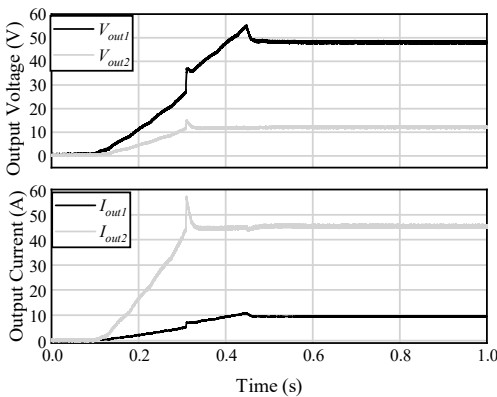

**Figure 7.** Dual port output waveforms.

**Table 2.** Measuring instruments.

| Instrument | Model Number |
| --- | --- |
| Oscilloscope | HDO6104A-MS (TELEDYNE) |
| Voltage differential probe | 700924 (YOKOGAWA) |
| Current probe (<30 A) | TCP312A (Tektronix) |
| Current probe (>30 A) | TCP303 (Tektronix) |
| Deskew calibration source | DCS025 (TELEDYNE) |

The voltage and current of both outputs were adjusted to constant values and steady operation was established within 0.5 s following power-up. The averages of the adjusted voltages $V_{out1}$ and $V_{out2}$ were 48.0 V and 12.0 V, respectively, as per the target values. The ripple voltages in the steady state, $V_{out1}$ and $V_{out2}$, were both ±0.6 V, whereas the ripple currents, $I_{out1}$ and $I_{out2}$, were ±0.2 A and ±0.8 A, respectively. The ripple current $I_{out2}$ was larger because the resolution of TCP303 used to measure $I_{out2}$ was lower than that of the TCP312A used to measure $I_{out1}$. These results indicate that the proposed control method can produce an output of 1 kW in an actual circuit.

The manner in which the voltage and current increased in this experiment was not a result of the proposed control method but rather, because of the voltage increase slew rate of the DC power supply ZX-1600H used in the verification. Furthermore, the control parameters of the proposed method (proportional gain of the time constant of the PI calculation part) were only optimized for the disturbance response, which is detailed later. This is because, if this circuit is used as designed, load changes will occur more frequently than power increases. The control of a circuit with a power supply voltage applied from the start is a topic for future research.

*3.3. Operation Points*

The circuit operation points used in the simulation and verification are illustrated in Figure 8. Six operation points of 154 W, 285 W, 456 W, 568 W, 740 W and 853 W were prepared on the $P_{out1}$ side using inductive resistance, whereas four operation points of 142 W, 268 W, 394 W and 499 W were prepared on the $P_{out2}$ side using non-inductive resistance. The tests and analysis were conducted at a total of 18 locations where the total output power $P_{total}$ ($P_{out1} + P_{out2}$) was less than 1000 W, which were defined as operation points. In the Section 3.4, Section 3.5, Section 3.6, Section 4.2, and Section 4.3, the numbers indicated in Figure 8 are used to denote the operation points.

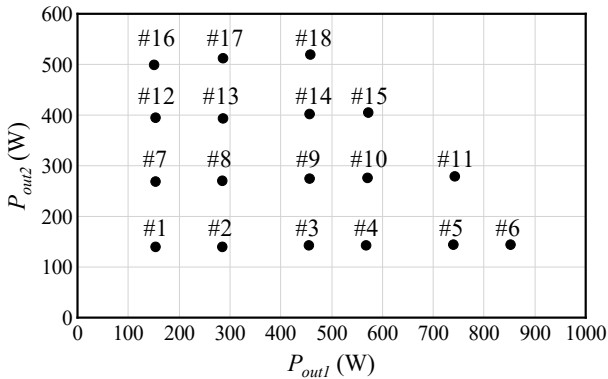

**Figure 8.** Measured operation points.

*3.4. Analysis of Steady Operation*

In the simulation, the waveform of each part of the circuit in steady operation was calculated and an analysis was performed on each operation point. PSIM (v. 12.04) was used for the simulation. Table 3 lists the specifications of simulation circuit. The values displayed in Table 3 were same as the experimental specifications listed in Table 1 except for the switching devices. All of the switching devices

and the transformer were ideal and there were no parasitic resistance, inductance and capacitance in the simulation circuit.

**Table 3.** Specifications of simulation circuit.

| Parameter | Value |
|---|---|
| $V_{in}$ | 300 V |
| $V_{out1}$ | 48 V |
| $V_{out2}$ | 12 V |
| $f_{swmin}$ | 50 kHz |
| $f_{swMAX}$ | 100 kHz |
| $f_{CLK}$ ($T_{CLK}$) | 200 MHz (5 ns) |
| $K_1$ | 2.5 |
| $K_2$ | 0.5 |
| $\tau_1$ | 3 μs |
| $\tau_2$ | 8 μs |
| $C_{in}$ | 330 μF |
| S1, S2, S3, S4 | Ideal devices |
| D11, D12, D13, D14, D21, D22, D23, D24 | Ideal devices |
| Transformer turn ratio | N1:N2 = 20:5 |
| $L_{sr}$ | 4.5 μH |
| $C_{sr}$ | 560 nF |
| $L_{out1}$ | 7.3 μH |
| $L_{out2}$ | 7.0 μH |
| $C_{out1}$ | 44 μF |
| $C_{out2}$ | 188 μF |

These analysis results refer to the operation mode of the secondary side circuit when $P_{out2}$ was maintained constant and $P_{out1}$ was changed (operation points #01, #02, #03, #04, #05 and #06) and when $P_{out1}$ was maintained constant and $P_{out2}$ was changed (operation points #01, #07, #12 and #16).

Figure 9 indicates the voltage and current direction of each part used to define the operation modes, whereas Table 4 lists the operation modes of the secondary side circuit in the half cycle in a steady state based on the voltage and current directions of Figure 9. As shown in Table 4, the operation modes were defined by the secondary side voltage and current of the transformer ($V_{txs}$ and $I_{txs}$), the current from the transformer to the diode bridge on the $V_{out1}$-side circuit ($I_{recA}$ and $I_{recB}$), the current of $L_{out1}$ ($I_{Lout1}$), the return current of the $V_{out1}$-side circuit ($I_{ret1}$), the currents of the series resonant filters ($I_{srA}$ and $I_{srB}$), the current of $L_{out2}$ ($I_{Lout2}$), the return current of the $V_{out2}$-side circuit ($I_{ret2}$) and the circulating current ($I_{cir}$) that flowed between the GND of both outputs. In Table 4, "+" represents the positive direction, "−" denotes the negative direction and "0" is the condition of no voltage applied or no current flowing.

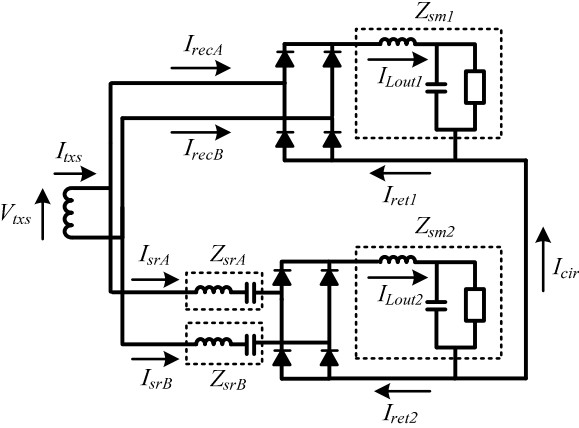

**Figure 9.** Definition of voltage and current directions.

**Table 4.** Definition of operation modes.

| Voltage/Current | 1 | 2 | 3 | 4 | 5 | 6 | 7 | 8 | 9 | 10 | 11 |
|---|---|---|---|---|---|---|---|---|---|---|---|
| $V_{txs}$ | + | + | + | + | + | 0 | 0 | 0 | 0 | 0 | 0 |
| $I_{txs}$ | + | + | + | + | + | + | + | + | − | − | − |
| $I_{recA}$ | + | + | + | + | + | + | + | + | + | + | + |
| $I_{recB}$ | − | − | − | − | − | − | − | − | − | − | 0 |
| $I_{Lout1}$ | + | + | + | + | + | + | + | + | + | + | + |
| $I_{ret1}$ | + | + | + | + | + | + | + | + | + | + | 0 |
| $I_{srA}$ | + | + | + | + | − | + | − | − | − | − | − |
| $I_{srB}$ | − | − | + | + | + | + | + | + | + | + | + |
| $I_{Lout2}$ | + | + | + | + | + | + | + | + | + | + | + |
| $I_{ret2}$ | + | + | + | 0 | + | + | + | + | + | + | + |
| $I_{cir}$ | − | + | + | + | + | + | + | − | + | − | − |

There were a total of 11 operation modes for the operation points of the analysis. Table 5 lists the possible operation modes for each operation point as indicated in Table 5, for each operation point, several modes were used and others were not, which created a large variety of current paths. As an example, Figure 10 depicts the simulation waveform of $V_{txs}$ and each current at operation points #04 and #12. The both waveforms of #04 and #12 are divided into 8 modes based on the definition of Table 4, respectively. The operation modes 3, 6 and 9 were not used at operation point #04 and the modes 4, 5 and 8 were not used at #12 as shown in Table 5. Therefore, it is difficult to explain the circuit operation by the 11 operation modes. However, by focusing on the charging condition of $L_{out1}$, the 11 operation modes could be categorized into three groups (A, B and C). The changes in $D_{tx}$ and $f_{tx}$ caused by the changes in the operation points can be explained by these three operation modes.

**Table 5.** Operation modes and measurement points.

| Operation Point | Operation Mode | | | | | | | | | | |
|---|---|---|---|---|---|---|---|---|---|---|---|
| | 1 | 2 | 3 | 4 | 5 | 6 | 7 | 8 | 9 | 10 | 11 |
| #01 | | ○ | ○ | | | | ○ | | | ○ | ○ |
| #02 | ○ | ○ | | ○ | | ○ | ○ | | | ○ | ○ |
| #03 | ○ | ○ | | ○ | | | ○ | | | ○ | ○ |
| #04 | ○ | ○ | | ○ | ○ | | ○ | ○ | | ○ | ○ |
| #05 | ○ | ○ | | ○ | ○ | | ○ | ○ | | ○ | ○ |
| #06 | ○ | ○ | | ○ | ○ | | ○ | ○ | | ○ | ○ |
| #07 | | ○ | ○ | | | ○ | ○ | | ○ | ○ | ○ |
| #12 | ○ | ○ | ○ | | | ○ | ○ | | ○ | ○ | ○ |
| #16 | ○ | ○ | ○ | | | | ○ | | ○ | ○ | ○ |

- Group A: $L_{out1}$ is charged (modes 1 to 5)

In this mode, $V_{txs}$ is positive. As an example, Figure 11a presents the current path in mode 2. The transformer voltage works as the voltage source and $L_{out1}$ is charged by the following path: Transformer → D11 → $Z_{sm1}$ → D14 → Transformer.

- Group B: $L_{out1}$ discharges (modes 6 to 10)

In this mode, $V_{txs}$ is zero and $I_{ret1}$ is flowing. As an example, the current path in mode 7 is illustrated in Figure 11b. When $L_{out1}$ is discharged, its discharge current returns along the following path: $Z_{sm1}$ → D14 → Transformer → D11 → $Z_{sm1}$.

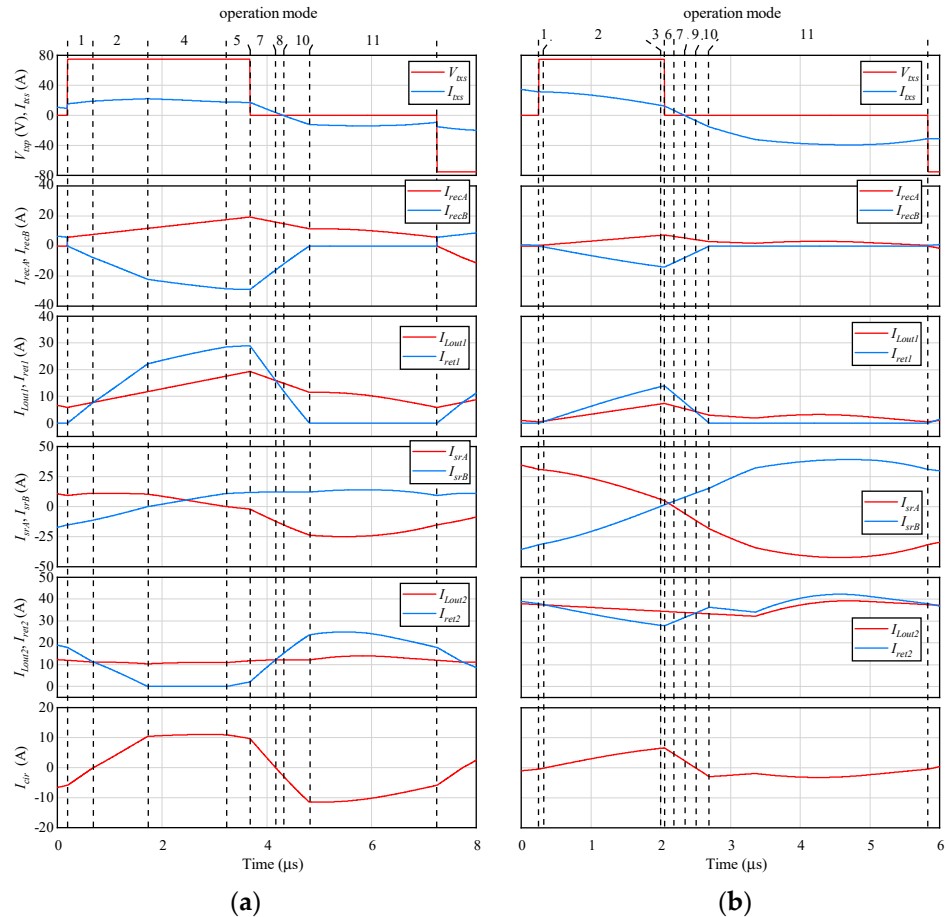

**Figure 10.** Waveforms of for half period: (**a**) measurement points #04; (**b**) measurement points #12.

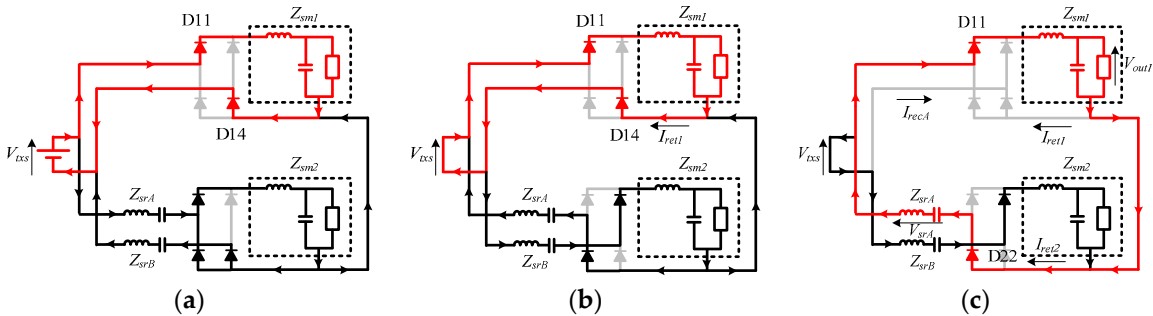

**Figure 11.** Current path of experimental circuit in each operation mode (the red line indicates the path related to $L_{out1}$): (**a**) Mode 2; (**b**) Mode 7; (**c**) Mode 10.

- Group C: The charge/discharge of $L_{out1}$ depends on the resonant filter voltage (mode 10)

Figure 11c depicts the current path of mode 10. In this mode, $I_{ret1}$ and $I_{recB}$ are zero. The return current from $Z_{sm1}$ becomes a circulating current ($I_{cir}$) and moves around the return current of the $V_{out2}$ side ($I_{ret2}$). This circulating current $I_{cir}$ flows because the GND of the $V_{out1}$-side and $V_{out2}$-side circuits are common and the return path of the $V_{out2}$-side circuit contains $Z_{srA}$ or $Z_{srB}$. The circulating current of this mode flows according to the state of $Z_{srA}$ and $Z_{srB}$. The current path involved in $L_{out1}$ is $Z_{sm1}$ → D22 →$Z_{srA}$ → D11 →$Z_{sm1}$. When the both-end voltage of $Z_{srA}$ ($V_{srA}$) exhibits the relationship of $V_{srA} > V_{out1}$, $L_{out1}$ is charged by $Z_{srA}$ and when $V_{srA} < V_{out1}$, $L_{out1}$ discharges.

Figure 12 presents the occupancy time and corresponding frequencies of these three modes at each operation point. When $P_{out2}$ was constant and $P_{out1}$ changed (Figure 12a), the time occupied by mode

A increased ($D_{tx}$ increased) and $f_{tx}$ decreased ($T_{tx}$ increased) as $P_{out1}$ increased. The increase in $D_{tx}$ was caused by the increase in $P_{out1}$ and the decrease in $f_{tx}$ occurred to prevent an increase in the power supply to the $V_{out2}$-side circuit caused by the increase in $D_{tx}$. However, when $P_{out1}$ was constant and $P_{out2}$ changed (Figure 12b), $P_{out2}$ increased as $T_{tx}$ decreased ($f_{tx}$ increased) but few changes occurred in the percentages of the three operation modes (changes in $D_{tx}$), regardless of the changes in $P_{out2}$. This indicates that the impedance change of the $V_{out2}$-side circuit caused by the frequency change played a dominant role in the changes in $P_{out2}$.

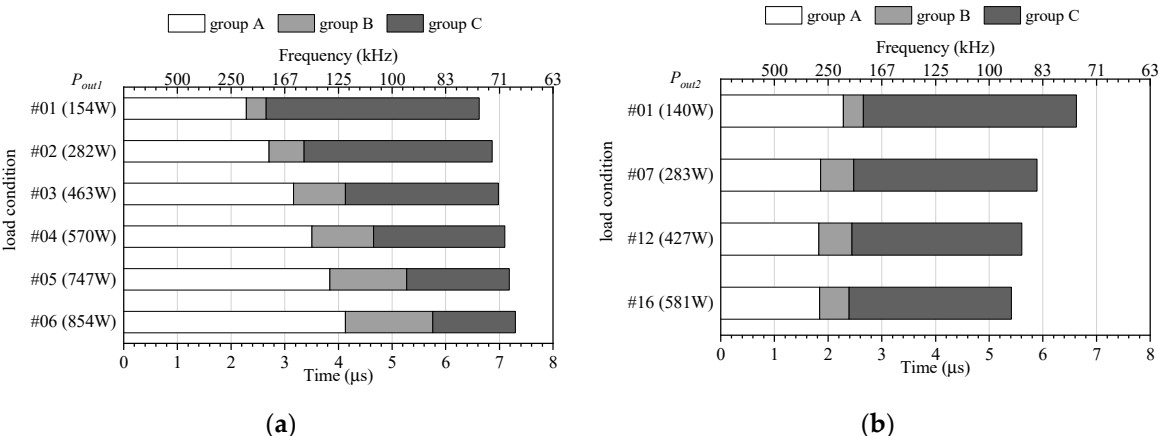

**Figure 12.** Occupied time and frequency of three operation modes: (**a**) $P_{out1}$ changes from 154 W to 853 W ($P_{out2}$ = 142 W); (**b**) $P_{out2}$ changes from 140 W to 581 W ($P_{out2}$ = 154 W).

### 3.5. Comparative Evaluation of Simulation and Measured Results

Figure 13 depicts the measured and simulation waveforms of each part at operation points #01, #06 and #16 in steady operation. From the top, Figure 13 presents the voltages of the primary and secondary sides of the transformer ($V_{txp}$ and $V_{txs}$), the currents of the primary and secondary sides of the transformer ($I_{txp}$ and $I_{txs}$), the both-end voltages of D11 and D12 ($V_{d11}$ and $V_{d12}$), the $L_{out1}$ current and output current ($I_{Lout1}$ and $I_{out1}$), the series resonant filter currents ($I_{srA}$ and $I_{srB}$), the series resonant filter voltages ($V_{srA}$ and $V_{srB}$), the both-end voltages of D21 and D22 ($V_{d21}$ and $V_{d22}$), the $L_{out2}$ current and output current ($I_{Lout2}$ and $I_{out2}$) and the circulating current ($I_{cir}$). According to Figure 13, the measured and simulation waveforms of #01, #06 and #16 were all very close, which confirms that the verification circuit operated as designed. The errors in $D_{tx}$ and $f_{tx}$, the voltage surge that only appeared in the measured waveforms and the ringing resulting from it were all caused by wiring resistance, parasitic inductance and floating capacitance, which were not included in the simulation.

Figures 14 and 15 present the relationships between $D_{tx}$ and $P_{out1}$ and $f_{tx}$ and $P_{out2}$, respectively, in steady operation for the measurement and simulation. In addition, the difference between the experimental results and the simulation results for Figures 14 and 15 are depicted in Figure 16. Figures 14–16 indicate the experimental tests are operated by larger $D_{tx}$ and lower $f_{tx}$ than the simulation tests under all load conditions. The reason for the larger $D_{tx}$ is that $D_{tx}$ compensates for $V_{out1}$ decreased by the conduction losses of parasitic resistance in the actual circuit and the reason for the lower $f_{tx}$ is that $f_{tx}$ controls increasing $V_{out2}$ with increasing $D_{tx}$. Although these errors appeared, it can be observed that the values of $D_{tx}$ and $f_{tx}$ at all operation points in steady operation were close to the simulation results. Therefore, the verification circuit also operated as per the simulation analysis at other operation points that are not shown in the waveforms of Figure 13. This is a further demonstration of the efficacy of the proposed control method.

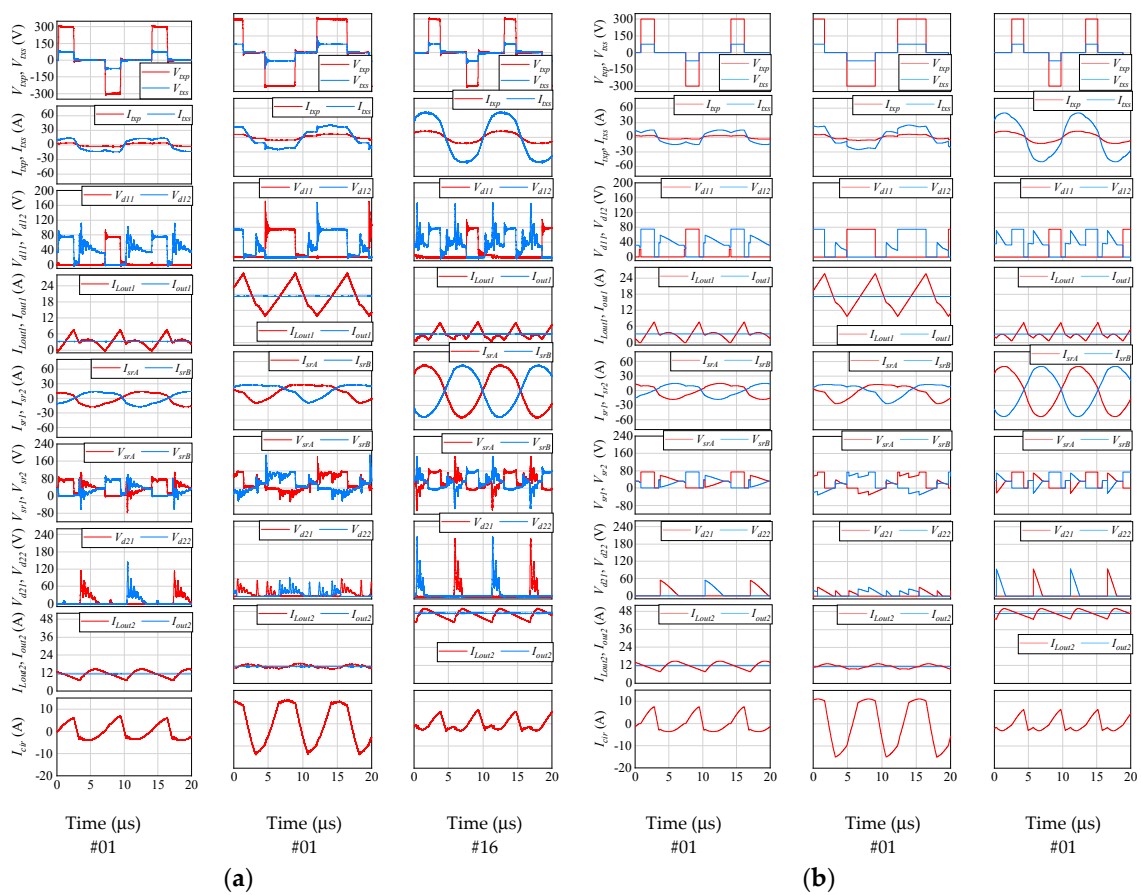

**Figure 13.** Waveforms of proposed circuit: (**a**) Experimental results; (**b**) Simulation results.

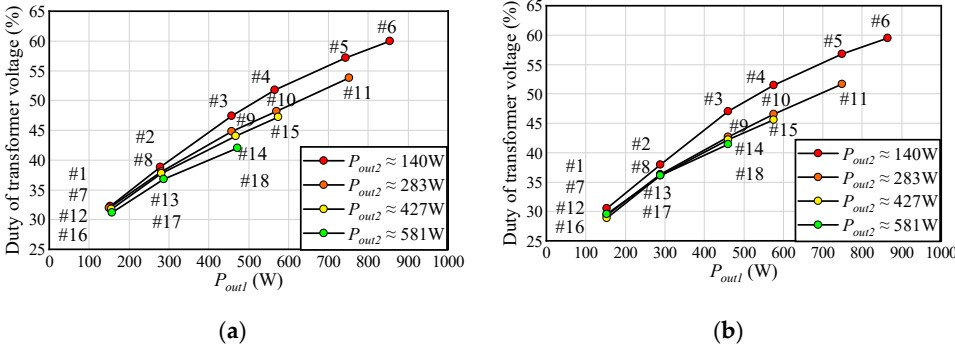

**Figure 14.** Results of $D_{tx}$ for each measurement point: (**a**) Experimental results; (**b**) Simulation results.

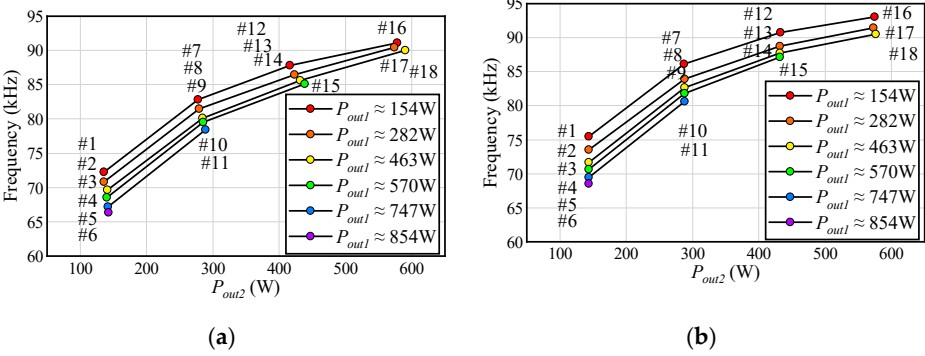

**Figure 15.** Results of $f_{tx}$ for each measurement point: (**a**) Experimental results; (**b**) Simulation results.

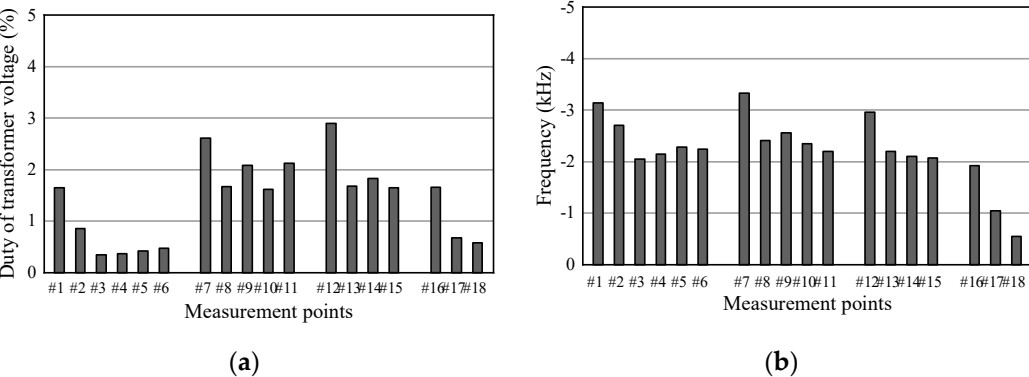

**Figure 16.** Results of subtract the simulation results from the experimental results for each measurement point: (**a**) results of $D_{tx}$; (**b**) results of $f_{tx}$.

### 3.6. Dynamic Characteristics

Figures 17 and 18 present the output voltage, measured current waveform and simulation results when $P_{out2}$ was maintained constant at 140 W and $P_{out1}$ was switched between 285 W and 568 W (#02 and #04 were switched) and when $P_{out1}$ was maintained constant at 286 W and $P_{out2}$ was switched between 270 W and 393 W (#08 and #13 were switched), respectively. Under any condition of Figures 17 and 18, within 8 ms after the output power was switched, both of the output voltages were adjusted to their respective values prior to the switch. In the measured waveforms of Figure 17, an overshoot with a peak value of around 2 V appeared in both output voltages $V_{out1}$ and $V_{out2}$ and an error of approximately 0.5 V occurred in the steady value of $V_{out1}$ in the measured waveform of Figure 18. However, overall, the measured and simulation waveforms were very close, confirming that the dynamic characteristics of the proposed control method were valid.

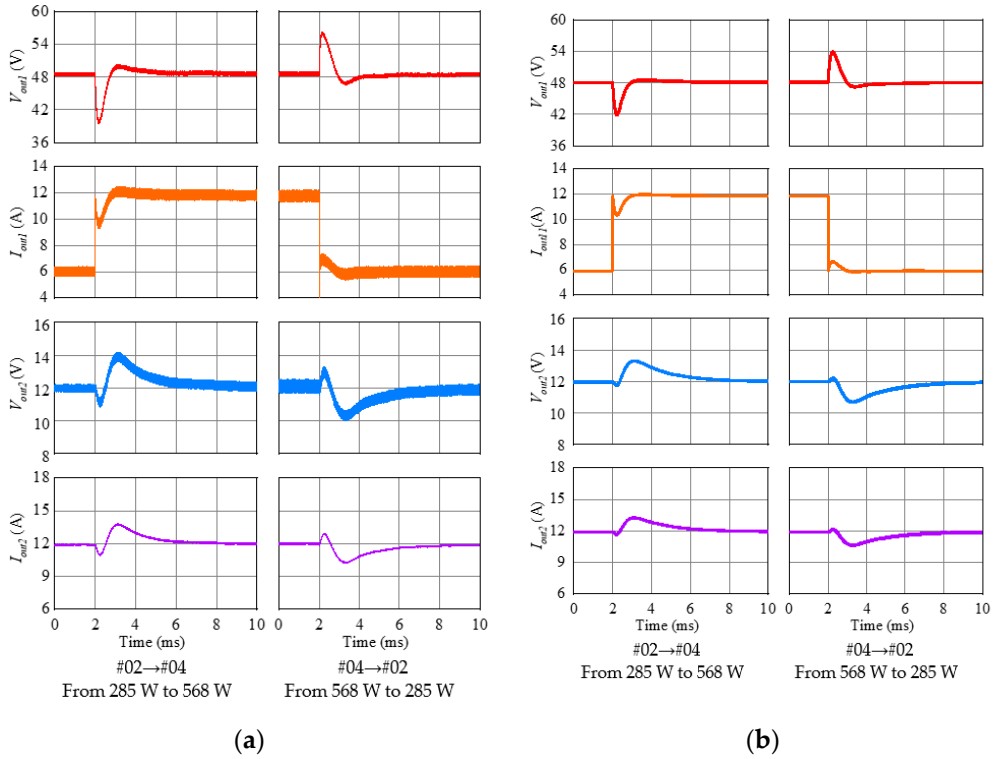

**Figure 17.** Results of $P_{out1}$ dynamic response test when $P_{out2}$ = 140 W: (**a**) Experimental results; (**b**) Simulation results.

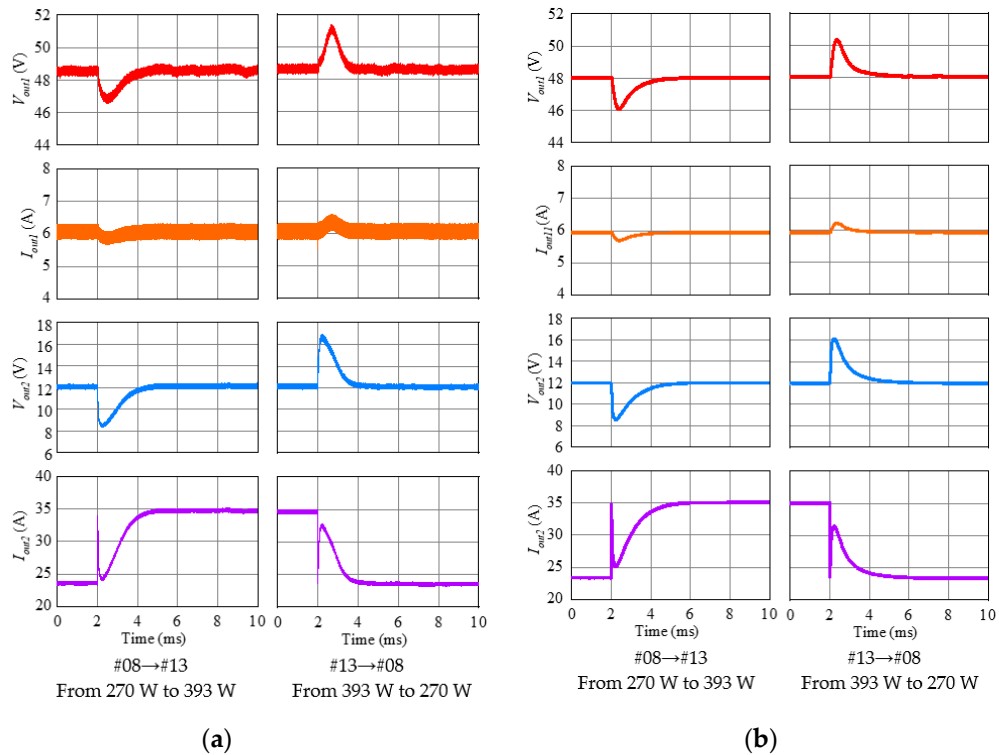

**Figure 18.** Results of $P_{out2}$ dynamic response test when $P_{out1}$ = 286 W: (**a**) Experimental results; (**b**) Simulation results.

## 4. Evaluation of Efficiency and Losses

### 4.1. Measurement Method

To evaluate the efficiency and losses of the verification circuit, the input/output power ($P_{in}$, $P_{out1}$ and $P_{out2}$) was measured with a WT1800 power analyzer. Furthermore, the voltage and current of the transformer primary side ($V_{txp}$, $I_{txp}$), the voltage and current of the transformer secondary side ($V_{txs}$, $I_{txs}$) and the voltage and current of the resonant filter of each series ($V_{srA}$, $V_{srB}$, $I_{srA}$, $I_{srB}$) were measured using the instruments listed in Table 2. The points of measurement are illustrated in Figure 19.

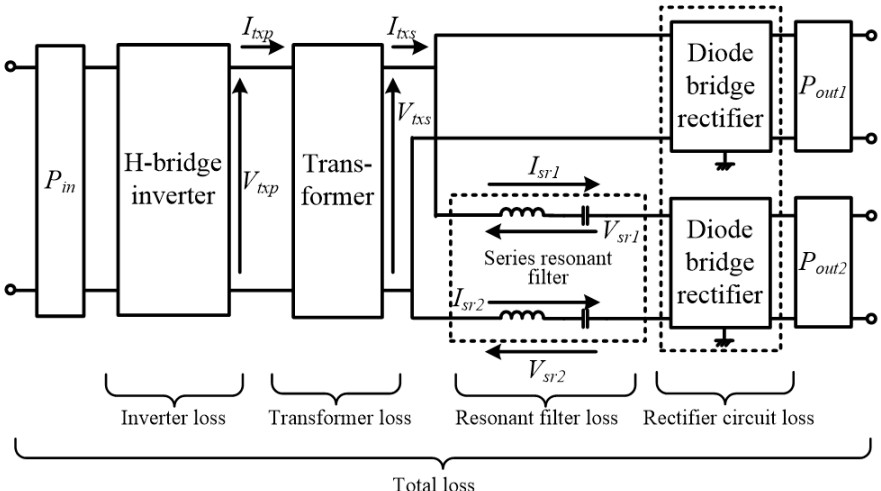

**Figure 19.** Points and loss positions of experimental test.

The efficiency $\eta$ of the measured power was calculated according to the following equation:

$$\eta = \frac{P_{out1} + P_{out2}}{P_{in}}. \tag{9}$$

For the measured voltage and current, the power of the primary side of the transformer ($P_{txp}$), power of the secondary side of the transformer ($P_{txs}$) and power of the resonant filters ($P_{srA}$ and $P_{srB}$) were calculated using the following equation:

$$P = \frac{1}{40T} \sum_{0}^{40T} v(t)i(t)\Delta T \quad n = 6, 7, 8, 9. \tag{10}$$

In the above, $n$ is the number of cycles contained in the 100 μs measured, $T$ is the length of a cycle, $v(t)$ and $i(t)$ are the measured voltage and current, respectively and $\Delta T$ is the time interval of the oscilloscope of 0.1 ns. Owing to the conditions of the measuring system, each series of data was obtained with three or four measurements, which could produce errors in the measurement results.

### 4.2. Efficiency Characteristics

Figure 20 depicts the efficiency at each operation point with $P_{out1}$ on the *x*-axis. The lowest efficiency of 68.3% was registered at point #16, where $P_{out1}$ was the minimum and $P_{out2}$ was the maximum. Meanwhile, the maximum efficiency was 88.9% at point #06, where $P_{out1}$ was the maximum and $P_{out2}$ was the minimum. When $P_{out2}$ was constant, the efficiency increased along with $P_{out1}$, regardless of the value of $P_{out2}$, indicating that the load losses caused by $P_{out2}$ and other fixed losses were larger than the load losses related to $P_{out1}$. However, when $P_{out1}$ was constant, the overall efficiency decreased as $P_{out2}$ increased, regardless of the value of $P_{out1}$. This indicates that the load losses related to $P_{out2}$ were larger than those caused by $P_{out1}$ and other fixed losses. Therefore, the load losses of $P_{out2}$ had a significant impact on the efficiency of this circuit.

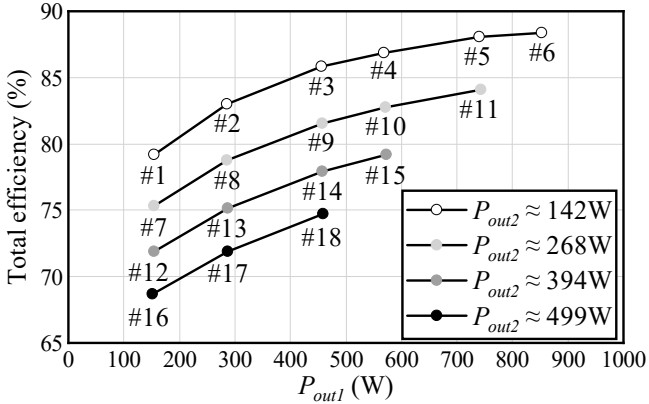

**Figure 20.** Results of efficiency measurements.

### 4.3. Loss Analysis

To analyze the loss points, loss separation was performed when the ratio of $P_{out1}$ to the total output power ($P_{out1} + P_{out2}$) was the maximum (#06), when the ratio of $P_{out2}$ was the maximum (#16) and under intermediate conditions (#10). For the power at each part obtained by the measurement and calculation, the losses at the inverter ($W_{inv}$), transformer ($W_{tx}$), series resonant filter ($W_{sr}$) and rectifier circuit ($W_{rec}$), as well as the total loss ($W_{total}$), were calculated using the equation shown in Table 6. The results are illustrated in Figure 21. The left axis indicates the losses and the right axis represents the ratio of $P_{out2}$ to the total output power.

**Table 6.** Loss calculations.

| Part | Symbol | Equation |
|------|--------|----------|
| Total loss | $W_{total}$ | $P_{in} - (P_{out1} + P_{out2})$ |
| Inverter loss | $W_{inv}$ | $P_{in} - P_{txp}$ |
| Transformer loss | $W_{tx}$ | $P_{txp} - P_{txs}$ |
| Resonant filter loss | $W_{sr}$ | $P_{r1} + P_{r2}$ |
| Rectifier circuit loss | $W_{rec}$ | $W_{total} - (W_{inv} + W_{tx} + W_{sr})$ |

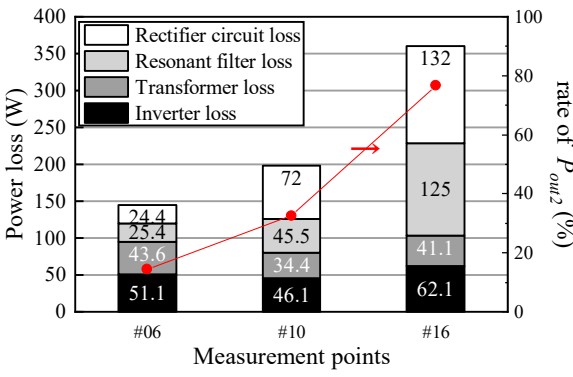

**Figure 21.** Loss analysis results.

As indicated in Figure 21, the $W_{total}$ values at the operation points were 144 W (#06), 198 W (#10) and 360 W (#16), whereas the ratios of $P_{out2}$ were 14.4% (#06), 32.5% (#10) and 76.7% (#16). Therefore, $W_{total}$ increased as the ratio of $P_{out2}$ increased. Figure 15 demonstrates that, as the ratio of $P_{out2}$ increased, $f_{tx}$ also increased and, simultaneously, the impedance of the $V_{out2}$ side decreased and the value of the filter current increased. As a result, the currents of the inverter, transformer and rectifier circuit increased, as did the conduction losses of each part. Moreover, the switching losses of the inverter and rectifier circuit increased, as did the core losses of the transformer and inductor. Therefore, as $f_{tx}$ increased, the losses of all parts increased; however, the fact that the increase ratio of $W_{sr}$ and $W_{rec}$ was larger than that of $W_{inv}$ and $W_{tx}$ and the total output power decreased as the ratio of $P_{out2}$ increased, indicates that there was little change in $W_{inv}$ and $W_{tx}$, whereas $W_{sr}$ and $W_{rec}$ increased at the three operation points, as illustrated in Figure 21.

Given that $W_{inv}$ and $W_{tx}$ at #06 (which exhibited the highest efficiency) accounted for more than half of $W_{total}$, it is possible to increase the maximum efficiency by reducing $W_{inv}$ and $W_{tx}$. It is necessary to select switching devices that suit the operation specifications to reduce $W_{inv}$, whereas a core with low loss should be selected and the core structure needs to be optimized to reduce $W_{tx}$. However, as the efficiency was the lowest when the ratio of $P_{out2}$ was the maximum (#16), $W_{sr}$ and $W_{rec}$, which were produced when the current of the $V_{out2}$-side circuit increased, limited the efficiency of the proposed circuit. Therefore, reducing these two values can effectively improve the overall efficiency of the circuit. Decreasing the peak value of the resonant filter current has been suggested as a possible means of reducing the $Q$ value of the series resonant filter but decreasing the $Q$ value means decreasing the sensitivity of the impedance changes to the frequency changes. In this case, it will be necessary to expand the operating frequency range to offset the amount of impedance change. However, if the range is expanded to the high-frequency side, an increased loss will be caused by the higher frequencies, as mentioned previously. Furthermore, if it is expanded to the low-frequency side, the problem of a volume increase in the core of the transformer and inductor will occur. Therefore, it is necessary to determine the optimal conditions.

## 5. Conclusions

This paper has presented a new method to control isolated single-input, dual-output DC/DC converters that are designed to be installed in EVs, PHVs and FCVs. The proposed method involves

dual voltage control by the duty cycle and frequency, which is realized by inserting parts with frequency-impedance characteristics into the circuit. To verify its efficacy, the proposed control method was applied to an actual circuit in which a series resonant filter was inserted in one of the diode bridge smoothing circuits that were connected in parallel to the secondary side of the transformer.

The results of the static characteristics indicate that two output voltages were adjusted to their respective target values 48 V and 12 V for the maximum output power 1037 W ($P_{out1}$ = 461 W, $P_{out2}$ = 576 W). The dynamic characteristics results show that when switching each output power, it recovered to a steady state within 8 ms after the switching of the operation point. The circuit behaviors at the different load conditions are analyzed by the simulation of the ideal circuit and the simulation waveforms were close to the experimental waveforms. These results demonstrated the validity of the proposed control method.

The efficiency of the circuit varied between 68.3% and 88.9%. The loss separation analysis indicated that the losses in the series resonant filter and diode bridge smoothing circuit account for 71% of the total loss and restricted the efficiency. The maximum efficiency of the circuit can be effectively increased by reducing the losses in the inverter and transformer because these losses constitute 66% of the total loss.

Our goal for future research is to examine the following points that were not covered in this study: controlling the circuit with a power supply voltage applied; improving the efficiency by considering modifications in the main circuit; and selecting the parts and designing the layout with the aim of miniaturization.

**Author Contributions:** Conceptualization, K.S., T.N.; methodology, K.S., Y.M. and T.N.; software, Y.M.; validation, Y.M., T.N.; formal analysis, Y.M.; investigation, Y.M.; data curation, Y.M.; writing—original draft preparation, Y.M.; writing—review and editing, Y.M., T.N.; supervision, T.N.; project administration, N.T., M.I. All authors have read and agreed to the published version of the manuscript.

**Funding:** This research received no external funding.

**Conflicts of Interest:** The authors declare no conflict of interest.

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
