# Peer review of "Control of Dual-Output DC/DC Converters Using Duty Cycle and Frequency"

_wevj, doi:10.3390/wevj11040072_

Round 1

Reviewer 1 Report

Presented here article described control system implementation in dual-output DC/DC converters and its test. The topic of the paper general fits to the journal topics. However the technical and scientific quality of the paper should be improved.  There are general remarks:

  • lack of novelty in relation to the existing works. This point requires a precise explanation,
  • no electrical schematic,
  • 5 is poorly marked,
  • How many tests were performed (statistics issue)?
  • provide error values when comparing the simulation and the real system,
  • describe the simulation conditions more precisely.

Reviewer 2 Report

This paper proposed a new method to control isolated single-input, dual-output DC/DC converters. To verify its efficacy, the proposed method was applied to an actual circuit. The experimental results of the new method showed the effectiveness of the new method. This paper is interesting to the readers. And the conclusion is innovative. The following comments are provided to the authors for improving this paper.

(1) It is recommended to add a schematic diagram of the experimental circuit setup, in addition to Figure 5.

(2) Could you please give more explanation according to Figure 10?

(3) It is suggested that the conclusion part can be divided into points to highlight the innovation of this paper.

Reviewer 3 Report

I reviewed the paper and please ask the authors to modify the paper based on these following suggestions:   1- please separate the Introduction section to 3 parts: research motivations, literature review, and research contributions. The novelty of the paper has to be explained in the contribution section.   2- For doing the simulation of Fig. 2, which software was used, and can you provide the exact characteristics of the circuit parameters in the separate table?!   3- Table 1 presented the specifications of the experimental and simulated parameters. Are they the same without any difference?! Because most of the time the experimental elements in the market are limited and we don’t have any options to replace them with the other parameters with various values. Please verify that all the parameters in both experimental and theoretical phases are similar.   4- I recommend to check and review following paper which was recently published about GaN inverter and add some helpful information in your introduction after reference [17]. Rahmani, F., Niknejad, P., Agarwal, T., & Barzegaran, M. (2020). Gallium Nitride Inverter Design with Compatible Snubber Circuits for Implementing Wireless Charging of Electric Vehicle Batteries. Machines8(3), 56.   5- I suggest to make Figures 15, 17, and 18 colorful and identify the graphs with the different colors.   6- In conclusion section, you should mention all your achieved theoretical and experimental numerical values. Please add more results, along with your method explanations. Thank you  

Round 2

Reviewer 1 Report

Thank you for your answers. In my opinion, the article in its current form is acceptable for publication.

Author Response

Thank you for your review. It is good training for me to check my manuscript from an objective perspective.